# What can linearized neural networks actually say about generalization?

**Guillermo Ortiz-Jiménez**
EPFL, Lausanne, Switzerland
`guillermo.ortizjimenez@epfl.ch`

**Seyed-Mohsen Moosavi-Dezfooli**
ETH Zurich, Zurich, Switzerland
`seyed.moosavi@inf.ethz.ch`

**Pascal Frossard**
EPFL, Lausanne, Switzerland
`pascal.frossard@epfl.ch`

## Abstract

For certain infinitely-wide neural networks, the neural tangent kernel (NTK) theory fully characterizes generalization, but for the networks used in practice, the empirical NTK only provides a rough first-order approximation. Still, a growing body of work keeps leveraging this approximation to successfully analyze important deep learning phenomena and design algorithms for new applications. In our work, we provide strong empirical evidence to determine the practical validity of such approximation by conducting a systematic comparison of the behavior of different neural networks and their linear approximations on different tasks. We show that the linear approximations can indeed rank the learning complexity of certain tasks for neural networks, even when they achieve very different performances. However, in contrast to what was previously reported, we discover that neural networks do not always perform better than their kernel approximations, and reveal that the performance gap heavily depends on architecture, dataset size and training task. We discover that networks overfit to these tasks mostly due to the evolution of their kernel during training, thus, revealing a new type of implicit bias.

## 1 Introduction

Due to their excellent practical performance, deep learning based methods are today the *de facto* standard for many visual and linguistic learning tasks. Nevertheless, despite this practical success, our theoretical understanding of how, and what, can neural networks learn is still in its infancy [1]. Recently, a growing body of work has started to explore the use of linear approximations to analyze deep networks, leading to the neural tangent kernel (NTK) framework [2].

The NTK framework is based on the observation that for certain initialization schemes, the infinite-width limit of many neural architectures can be exactly characterized using kernel tools [2]. This reduces key questions in deep learning theory to the study of linear methods and convex functional analysis, for which a rich set of theories exist [3]. This intuitive approach has been proved very fertile, leading to important results in generalization and optimization of very wide networks [4–9].

The NTK theory, however, can only fully describe certain infinitely wide neural networks, and for the narrow architectures used in practice, it only provides a first-order approximation of their training dynamics (see Fig. 1). Despite these limitations, the intuitiveness of the NTK, which allows to use a powerful set of theoretical tools to exploit it, has led to a rapid increase in the amount of research that successfully leverages the NTK in applications, such as predicting generalization [10] and training speed [11], explaining certain inductive biases [12–15] or designing new classifiers [16, 17].

35th Conference on Neural Information Processing Systems (NeurIPS 2021).

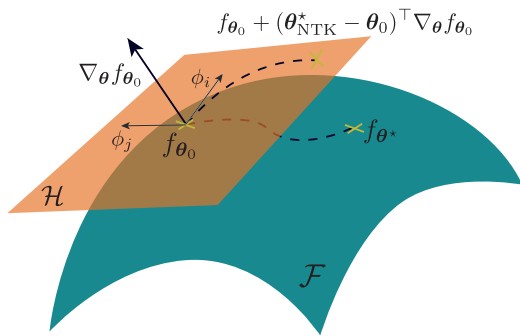

Figure 1: Conceptual illustration of the NTK approximation (see Sec 2). The empirical NTK defines a linear function space tangent $\mathcal{H}$ to the non-linear function space $\mathcal{F}$ defined by the network. In the limit of infinite width, the neural network space loses its curvature and coincides with the tangent space. Training a linearized network restricts the optimization trajectory to lie in the tangent space.

Recent reports, however, have started questioning the effectiveness of this approach, as one can find multiple examples in which kernel methods are provably outperformed by neural networks [18–20]. Most importantly, it has been observed empirically that linearized models – computed using a first-order Taylor expansion *around the initialization* of a neural network – perform much worse than the networks they approximate on standard image recognition benchmarks [21]; a phenomenon that has been coined as the *non-linear advantage*. However, it has also been observed that, if one linearizes the network at a later stage of training, the non-linear advantage is greatly reduced. The reasons behind this phenomenon are poorly understood, yet they are key to explain the success of deep learning.

Building from these observations, we delve deeper into the source of the non-linear advantage, trying to understand why previous work could successfully leverage the NTK in some applications. In particular, we shed new light on the question: *When can the NTK approximation be used to predict generalization, and what does it actually say about it?* We propose, for the first time, to empirically study this problem from the perspective of the characteristics of the training labels. To that end, we conduct a systematic analysis comparing the performance of different neural network architectures with their kernelized versions, on several problems with the same data support, but different labels. Doing so, we identify the alignment of the target function with the NTK as a key quantity governing important aspects of generalization in deep learning. Namely, one can rank the learning complexity of solving certain tasks with deep networks according to their kernel alignment.

We, then, study the evolution of the alignment during training to see its influence on the inductive bias. Prior work had shown that, during optimization, deep networks significantly increase their alignment with the target function, and this had been strongly conjectured to be positive for generalization [22–24]. In contrast, in this work, we offer a more nuanced view of this phenomenon, and show that it does not always have a positive effect for generalization. In fact, we provide multiple concrete examples where deep networks exhibit a *non-linear disadvantage* compared to their kernel approximations.

The main contributions of our work are:

- We show that the alignment with the empirical NTK at initialization can provide a good measure of relative learning complexity in deep learning for a diverse set of tasks.
- We use this fact to shed new light on the directional inductive bias of most convolutional neural networks, as this alignment can be used to identify neural anisotropy directions [25].
- Moreover, we identify a set of non-trivial tasks in which neural networks perform worse than their linearized approximations, and show this is due to their non-linear dynamics.
- We, hence, provide a fine-grained analysis of the evolution of the kernel during training, and show that the NTK rotates mostly in a single axis. This mechanism is responsible for the rapid convergence of neural networks to the training labels, but interestingly we find that it can sometimes hurt generalization, depending on the target task.

We see our empirical findings as an important step forward in our understanding of deep learning. Our work paves the way for new research avenues based on our newly observed phenomena, but it also provides a fresh perspective to understand how to use the NTK approximation in several applications.

## 2 Preliminaries

Let $f_{\boldsymbol{\theta}} : \mathbb{R}^d \to \mathbb{R}$ denote a neural network parameterized by a set of weights $\boldsymbol{\theta} \in \mathbb{R}^n$. Without loss of generality, but to reduce the computational load, in this work, we only consider the binary classification setting, where the objective is to learn an underlying target function $f : \mathbb{R}^d \to \{\pm 1\}$ by training the weights to minimize the empirical risk $\hat{\mathcal{R}}(f_{\boldsymbol{\theta}}) = \frac{1}{m} \sum_{i=1}^{m} \mathbb{1}_{\text{sign}(f_{\boldsymbol{\theta}}(\boldsymbol{x}_i) \neq f(\boldsymbol{x}_i))}$ over a finite set of i.i.d. training data $\mathcal{S} = \{(\boldsymbol{x}_i, f(\boldsymbol{x}_i))\}_{i=1}^{m}$ from an underlying distribution $\mathcal{D}$. We broadly say that a model *generalizes* whenever it achieves a low expected risk $\mathcal{R}(f_{\boldsymbol{\theta}}) = \mathbb{E}_{\boldsymbol{x} \sim \mathcal{D}}[\mathbb{1}_{\text{sign}(f_{\boldsymbol{\theta}}(\boldsymbol{x}) \neq f(\boldsymbol{x}))}]$.

In a small neighborhood around the weight initialization $\boldsymbol{\theta}_0$, a neural network can be approximated using a first-order Taylor expansion (see Fig. 1)

$$f_{\boldsymbol{\theta}}(\boldsymbol{x}) \approx \hat{f}_{\boldsymbol{\theta}}(\boldsymbol{x}; \boldsymbol{\theta}_0) = f_{\boldsymbol{\theta}_0}(\boldsymbol{x}) + (\boldsymbol{\theta} - \boldsymbol{\theta}_0)^\top \nabla_{\boldsymbol{\theta}} f_{\boldsymbol{\theta}_0}(\boldsymbol{x}), \tag{1}$$

where $\nabla_{\boldsymbol{\theta}} f_{\boldsymbol{\theta}_0}(\boldsymbol{x}) \in \mathbb{R}^n$ denotes the Jacobian of the network with respect to the parameters evaluated at $\boldsymbol{\theta}_0$. Here, the model $\hat{f}_{\boldsymbol{\theta}}$ represents a *linearized network* which maps weight vectors to functions living in a reproducible kernel Hilbert space (RKHS) $\mathcal{H} \subseteq L_2(\mathbb{R}^d)$, determined by the empirical NTK [4] at $\boldsymbol{\theta}_0$, $\boldsymbol{\Theta}_{\boldsymbol{\theta}_0}(\boldsymbol{x}, \boldsymbol{x}') = \langle \nabla_{\boldsymbol{\theta}} f_{\boldsymbol{\theta}_0}(\boldsymbol{x}), \nabla_{\boldsymbol{\theta}} f_{\boldsymbol{\theta}_0}(\boldsymbol{x}') \rangle$. Unless stated otherwise, we will generally drop the dependency on $\boldsymbol{\theta}_0$ and use $\boldsymbol{\Theta}$ to refer to the NTK at initialization.

In most contexts, the NTK evolves during training by following the trajectory of the network Jacobian $\nabla_{\boldsymbol{\theta}} f_{\boldsymbol{\theta}_t}$ computed at a checkpoint $\boldsymbol{\theta}_t$ (see Fig. 1). Remarkably, however, it was recently discovered that for certain types of initialization, and in the limit of infinite network-width, the approximation in (1) is exact, and the NTK is constant throughout training [2]. In this regime, one can, then, provide generalization guarantees for neural networks using generalization bounds from kernel methods, and show [26], for instance, that with high probability

$$\mathcal{R}(f^\star) \leq \hat{\mathcal{R}}(f^\star) + \mathcal{O}\left(\sqrt{\frac{\|f\|_{\boldsymbol{\Theta}}^2 \mathbb{E}_{\boldsymbol{x}}[\boldsymbol{\Theta}(\boldsymbol{x}, \boldsymbol{x})]}{m}}\right), \tag{2}$$

where $f^\star = \arg\min_{h \in \mathcal{H}} \hat{\mathcal{R}}(h) + \mu \|h\|_{\boldsymbol{\Theta}}^2$, with $\mu > 0$ denoting a regularization constant, and $\|f\|_{\boldsymbol{\Theta}}^2$ being the RKHS norm of the target function, which for positive definite kernels can be computed as

$$\|f\|_{\boldsymbol{\Theta}}^2 = \sum_{j=1}^{\infty} \frac{1}{\lambda_j} \left(\mathbb{E}_{\boldsymbol{x} \sim \mathcal{D}}[\phi_j(\boldsymbol{x}) f(\boldsymbol{x})]\right)^2. \tag{3}$$

Here, the couples $\{(\lambda_j, \phi_j)\}_{j=1}^{\infty}$ denote the eigenvalue-eigenfunction pairs, in order of decreasing eigenvalues, of the Mercer's decomposition of the kernel, i.e., $\boldsymbol{\Theta}(\boldsymbol{x}, \boldsymbol{x}') = \sum_{j=1}^{\infty} \lambda_j \phi_j(\boldsymbol{x}) \phi_j(\boldsymbol{x}')$. This means that, in kernel regimes, the difference between the empirical and the expected risk is smaller when training on target functions with a lower RKHS norm. That is, whose projection on the eigenfunctions of the kernel is mostly concentrated along its largest eigenvalues. One can then see the RKHS norm as a metric that ranks the complexity of learning different targets using the kernel $\boldsymbol{\Theta}$.

Estimating (3) in practice is challenging as it requires access to the smallest eigenvalues of the kernel. However, one can use the following lemma to compute a more tractable bound of the RKHS norm, which shows that a high target-kernel alignment is a good proxy for a small RKHS norm.

**Lemma 1.** *Let $\alpha(f) = \mathbb{E}_{\boldsymbol{x}, \boldsymbol{x}' \sim \mathcal{D}}[f(\boldsymbol{x}) \boldsymbol{\Theta}(\boldsymbol{x}, \boldsymbol{x}') f(\boldsymbol{x}')]$ denote the alignment of the target $f \in \mathcal{H}$ with the kernel $\boldsymbol{\Theta}$. Then $\|f\|_{\boldsymbol{\Theta}}^2 \geq \|f\|_2^4 / \alpha(f)$. Moreover, for the NTK, $\alpha(f) = \|\mathbb{E}_{\boldsymbol{x}}[f(\boldsymbol{x}) \nabla_{\boldsymbol{\theta}} f_{\boldsymbol{\theta}_0}(\boldsymbol{x})]\|_2^2$.*

*Proof* See Appendix.

At this point, it is important to highlight that in most practical applications we do not deal with infinitely-wide networks, and hence (2) can only be regarded as a learning guarantee for linearized models, i.e. $\hat{f}_{\boldsymbol{\theta}}$. Furthermore, from now on, we will interchangeably use the terms NTK and empirical NTK to simply refer to the finite-width kernels derived from (1). In our experiments, we compute those using the `neural_tangents` library [27] built on top of the JAX framework [28], which we also use to generate the linearized models.

Similarly, as it is commonly done in the kernel literature, we will use the eigenvectors of the Gram matrix to approximate the values of the eigenfunctions $\phi_j(\boldsymbol{x})$ over a finite dataset. We will also use the terms eigenvector and eigenfunction interchangeably. In particular, we will use $\boldsymbol{\Phi} \in \mathbb{R}^{m \times m}$ to denote the matrix containing the $j$th Gram eigenvector $\boldsymbol{\phi}_j \in \mathbb{R}^m$ in its $j$th row, where the rows are ordered according to the vector of decreasing eigenvalues $\boldsymbol{\lambda} \in \mathbb{R}_+^m$.

# 3 Linearized models can predict relative task complexity for deep networks

A growing body of work is using the linear approximation of neural networks as kernel methods to analyze and build novel algorithms. Meanwhile, recent reports, both theoretical and empirical, have started to question if the NTK approximation can really tell anything useful about generalization for finite-width networks. In this section, we try to demystify some of these confusions and aim to shed light on the question: *What can the empirical NTK actually predict about generalization?*

To that end, we conduct a systematic study with different neural networks and their linearized approximations given by (1), which we train to solve a structured array of predictive tasks with different complexity. Our results indicate that for many problems the linear models and the deep networks do agree in the way they *order* the complexity of learning certain tasks, even if their performance on the same problems can greatly differ. This explains why the NTK approximation can be used in applications where the main goal is to *just* predict the relative difficulty of different tasks.

## 3.1 Learning NTK eigenfunctions

In kernel theory, the sample and optimization complexity required to learn a given function is normally bounded by its kernel norm [3], which intuitively measures the alignment of the target function with the eigenfunctions of the kernel. The eigenfunctions themselves, thus, represent a natural set of target functions with increasingly high learning complexity – according to the increasing value of their associated eigenvalues – for kernel methods. Since our goal is to find if the kernel approximation can indeed predict generalization for neural networks, we evaluate the performance of these networks when learning the eigenfunctions of their NTKs.

In particular, we generate a sequence of datasets constructed using the standard CIFAR10 [29] samples, which we label using different binarized versions of the NTK eigenfunctions. That is, to every sample $x$ in CIFAR10 we assign it the label $\text{sign}(\phi_j(x))$, where $\phi_j$ represents the $j$th eigenfunction of the NTK at initialization (see Sec. 2). In this construction, the choice of CIFAR10 as supporting distribution makes our experiments close to real settings which might be conditioned by low dimensional structures in the data manifold [19, 23]; while the choice of eigenfunctions as targets guarantees a progressive increase in complexity, *at least*, for the linearized networks. Specifically, for $\phi_j$ the alignment is given by $\alpha(\phi_j) = \lambda_j$ (see Appendix).

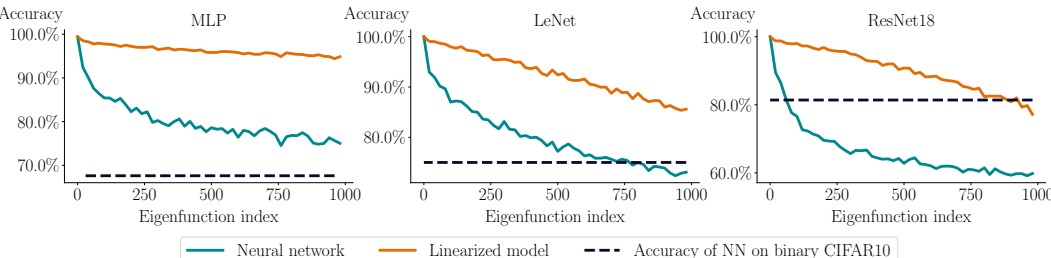

Figure 2: Validation accuracy of different neural network architectures and their linearizations when trained on binarized eigenfunctions of the NTK at initialization, i.e., $x \mapsto \text{sign}(\phi_j(x))$. As a baseline, we also provide the accuracies on CIFAR2 (see Sec. 4).

We train different neural network architectures – selected to cover the spectrum of small to large models [30–32] – and their linearized models given by (1). Unless stated otherwise, we always use the same standard training procedure consisting of the use of stochastic gradient descent (SGD) to optimize a logistic loss, with a decaying learning rate starting at $0.05$ and momentum set to $0.9$. The values of our metrics are reported after $100$ epochs of training[1].

Fig. 2 summarizes the main results of our experiments[2]. Here, we can clearly see how the validation accuracy of networks trained to predict targets aligned with $\{\phi_j\}_{j=1}^{\infty}$ progressively drops with decreasing eigenvalues for both linearized models – as predicted by the theory – as well as for neural networks. Similarly, Fig. 3 shows how the training dynamics of these networks also correlate with

---

[1]Our code can be found at `https://github.com/gortizji/linearized-networks`.

[2]Results with equivalent findings for other training schemes and datasets can be found in the Appendix.

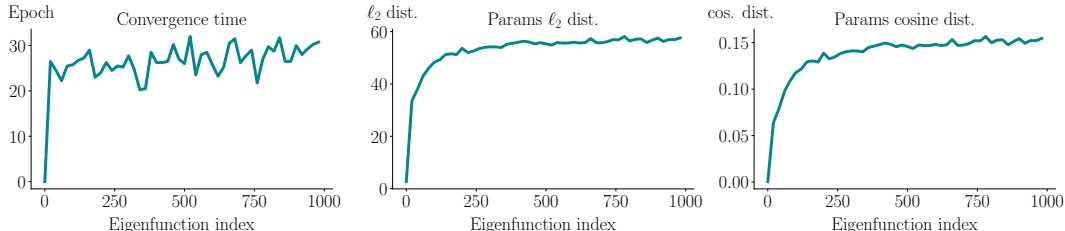

Figure 3: Correlation of different training metrics with the index of the eigenfunction the network is trained on. Plots show the number of training iterations taken by the network to achieve a 0.01 training loss, and the $\ell_2$ and cosine distances between initialization and final parameters for a ResNet18 trained on the binarized eigenfunctions of the NTK at initialization.

eigenfunction index. Specifically, we see that networks take more time to fit eigenfunctions associated to smaller eigenvalues, and need to travel a larger distance in the weight space to do so.

Overall, our observations reveal that sorting tasks based on their alignment with the NTK is a good predictor of learning complexity both for linear models and neural networks. Interestingly, however, we can also observe large performance gaps between the models. Indeed, even if the networks and the kernels agree on which eigenfunctions are harder to learn, the kernel methods perform comparatively much better. This clearly differs from what was previously observed for other tasks [18–21], and hence, highlights that the existence of a *non-linear advantage* is not always certain.

## 3.2 Learning linear predictors

The NTK eigenfunctions are one example of a canonical set of tasks with increasing hardness for kernel methods, whose learning complexity for neural networks follows the same order. However, could there be more examples? And, are the previously observed correlations useful to predict other generalization phenomena? In order, to answer these questions, we propose to analyze another set of problems, but this time using a sequence of targets of increasing complexity for neural networks.

In this sense, it has recently been observed that, for convolutional neural networks (CNNs), the set of linear predictors – i.e., hyperplanes separating two distributions – represents a function class with a wide range of learning complexities among its elements [25]. In particular, it has been confirmed empirically that it is possible to rank the complexity for a neural network to learn different linearly separable tasks based only on its neural anisotropy directions (NADs).

**Definition** (Neural anisotropy directions). *The NADs of a neural network are the ordered sequence of orthonormal vectors $\{v_j\}_{j=1}^d$ which form a full basis of the input space and whose order is determined by the sample complexity required to learn the linear predictors $\{x \mapsto \text{sign}(v_j^\top x)\}_{j=1}^d$.*

In [25], the authors provided several heuristics to compute the NADs of a neural network. However, we now provide a new, more principled, interpretation of the NADs, showing one can also obtain this sequence using a kernel approximation. To that end, we will make use of the following theorem.

**Theorem 1.** *Let $u \in \mathbb{S}^{d-1}$ be a unitary vector that parameterizes a linear predictor $g_u(x) = u^\top x$, and let $x \sim \mathcal{N}(0, I)$. The alignment of $g_u$ with $\Theta$ is given by*

$$\alpha(g_u) = \left\| \mathbb{E}_x \left[ \nabla_{x,\theta}^2 f_{\theta_0}(x) \right] u \right\|_2^2, \tag{4}$$

*where $\nabla_{x,\theta}^2 f_{\theta_0}(x) \in \mathbb{R}^{n \times d}$ denotes the derivative of $f_\theta$ with respect to the weights and the input.*

*Proof* See Appendix.

Theorem 1 gives an alternative method to compute NADs. Indeed, in the kernel regime, the NADs are simply the right singular vectors of the matrix of mixed-derivatives $\mathbb{E}_x \nabla_{x,\theta}^2 f_{\theta_0}(x)$ of the network[3]. Note however, that this interpretation is just based on an approximation, and hence there is no *explicit* guarantee that these NADs will capture the direcional inductive bias of deep networks. Our experiments show otherwise, as they reveal that CNNs actually rank the learning complexity of different linear predictors in a way compatible with Theorem 1.

---

[3]All predictors have the same $L_2$ norm. Hence, their alignment is inversely proportional to their kernel norm.

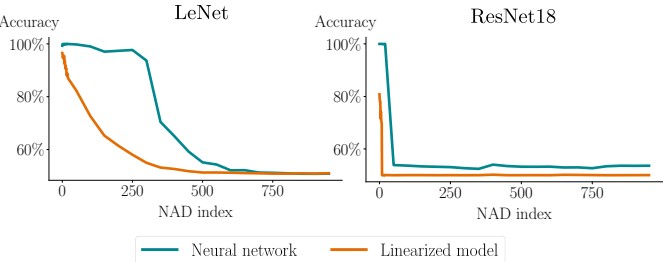
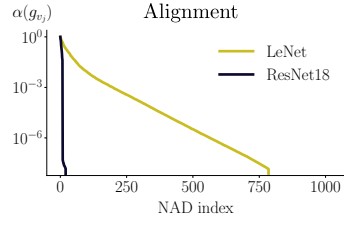

Figure 4: (Left) Performance comparison of different neural network architectures with their linearizations when learning linear target functions aligned with increasing NADs, i.e., $\boldsymbol{x} \mapsto \text{sign}(\boldsymbol{v}^\top \boldsymbol{x})$. (Right) Predicted value of the alignment of the predictors with $\boldsymbol{\Theta}$, i.e., $\alpha(g_{\boldsymbol{v}_j})$ (see Theorem 1)

Indeed, as shown in Fig. 4, when trained to classify a set of linearly separable datasets[4], aligned with the NTK-predicted NADs, CNNs perform better on those predictors with a higher kernel alignment (i.e., corresponding to the first NADs) than on those with a lower one (i.e., later NADs). The fact that NADs can be explained using kernel theory constitutes another clear example that theory derived from a naïve linear expansion of a neural network can sometimes capture important trends in the inductive bias of deep networks; even when we observe a clear performance gap between linear and non-linear models. Surprisingly, neural networks exhibit a strong *non-linear advantage* on these tasks, even though these NADs were explicitly constructed to be well-aligned with the linear models.

On the other hand, the fact that the empirical NTK of standard networks presents such strong directional bias is remarkable on its own, as it reveals important structure of the underlying architectures. Interestingly, recent theoretical studies [33] have found that the standard rotational invariance assumption in kernel theory [3] might be too restrictive to explain generalization in many settings. Hence, showing that the kernels of neural networks have a strong rotational variance, clearly strengthens the link between the study of these kernels and deep learning theory.

Overall, our results explain why previous heuristics that used the NTK to rank the complexity of learning certain tasks [10, 11, 13] were successful in doing so. Specifically, by systematically evaluating the performance of neural networks on tasks of increasing complexity for their linearized approximations, we have observed that the non-linear dynamics on these networks do not change the way in which they sort the complexity of these problems. However, we should not forget that the differences in performance between the neural networks and their approximation are very significant and whether they favor or not neural networks depends on the task.

# 4 Sources of the non-linear (dis)advantage

In this section, we study in more detail the mechanisms that separate neural networks from their linear approximations and that lead to their significant performance differences. Specifically, we will show that there are important nuances involved in the comparison of linear and non-linear models, which depend on number of training samples, architecture and target task.

To shed more light on these complex relations, we will conduct a fine-grained analysis of the dynamics of neural networks and study the evolution of their empirical NTK. We will show that the kernel dynamics can explain why neural networks converge much faster than kernel methods, even though this rapid adaptation can sometimes be imperfect and lead the networks to overfit.

## 4.1 The non-linear advantage depends on the sample size

As we have seen, there exist multiple problems in which neural networks perform significantly better than their linearized approximations (see Sec. 3.2), but also others where they do not (see Sec. 3.1). We now show, however, that the magnitude of these differences is influenced by the training set size.

We can illustrate this phenomenon by training several neural networks to predict some semantically-meaningful labels of an image dataset. In particular, and for consistency with the previous two-class

---

[4]Full details of the experiment can be found in the Appendix.

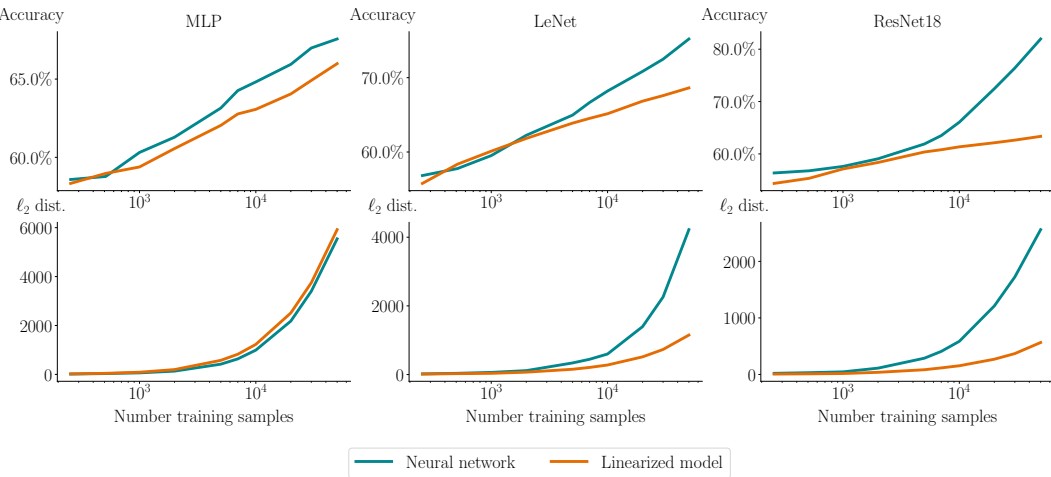

Figure 5: Comparison of test accuracy (top) and parameter distance to initialization (bottom) between neural networks and their linear approximations trained on CIFAR2 with different training set sizes. Plots show average over five different random seeds and when the test accuracy has saturated.

examples, we deal with a binary version of CIFAR10 and assign label $+1$ to all samples from the first five classes in the dataset, and label $-1$ to the rest. We will refer to this dataset as CIFAR2. Indeed, as seen in Fig. 5, some neural networks exhibit a large non-linear advantage on this task, but this advantage mostly appears when training on larger datasets. This phenomenon suggests that the inductive bias that boosts neural networks' performance involves an important element of scale.

One can intuitively understand this behavior by analyzing the distance traveled by the parameters during optimization (see bottom row of Fig. 5). Indeed, for smaller training set sizes, the networks can find solutions that fit the training data closer to their initialization more easily[5] As a result, the error incurred by the linear approximation in these cases is smaller. This can explain why there are no significant performance gaps between NTK-based models and neural networks for small-data [16], and it also highlights the strength of the linear approximation in this regime.

## 4.2 The kernel rotates in a single axis

So far we have mostly analyzed results dealing with linear expansions around the weight initialization $\boldsymbol{\theta}_0$. However, recent empirical studies have argued that linearizing at later stages of training induces smaller approximation errors [21], suggesting that the NTK dynamics can better explain the final training behavior of a neural network. To the best of our knowledge, this phenomenon is still poorly understood, mostly because it hinges on understanding the non-linear dynamics of deep networks. We now show, however, that understanding the way the spectrum of the NTK evolves during training can provide important insights into these dynamics.

To that end, we first analyze the evolution of the principal components of the empirical NTK in relation to the target function. Specifically, let $\boldsymbol{\Phi}_t$ denote the matrix of first $K$ eigenvectors of the Gram matrix of $\boldsymbol{\Theta}_t$ obtained by linearizing the network after $t$ epochs of training, and let $\boldsymbol{y} \in \mathbb{R}^m$ be the vector of training labels. In Sec. 3.1, we have seen that both neural networks and their linear approximations perform better on targets aligned with the first eigenfunctions of $\boldsymbol{\Theta}_0$. We propose, therefore, to track the energy concentration $\|\boldsymbol{\Phi}_t \boldsymbol{y}\|_2 / \|\boldsymbol{y}\|_2$ of the labels onto the first eigenfunctions with the aim to identify a significant transformation of the principal eigenspace $\mathrm{span}(\boldsymbol{\Phi}_t)$.

Fig. 6 shows the result of this procedure applied to a CNN trained to classify CIFAR2. Strikingly, the amount of energy of the target function that is concentrated on the $K = 50$ first eigenfunctions of the NTK significantly grows during training. This is a heavily non-linear phenomenon – by definition

---

[5]Note that linear and non-linear models achieve their maximum test accuracy in roughly the same number of epochs regardless of the training set size. This contrasts with the dynamics of the training loss, for which the linear models take significantly more iterations to perfectly fit the training data than non-linear ones. In this sense, the results of Fig. 5 present a snapshot taken when both networks approximately achieve their maximum generalization performance.

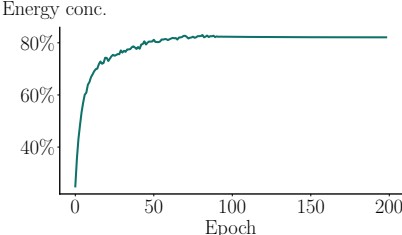
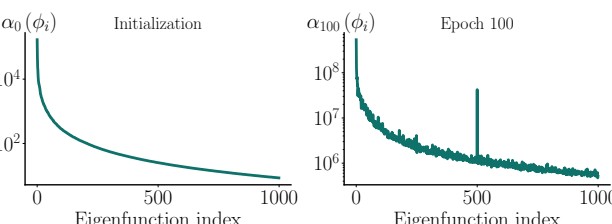

Figure 6: Energy concentration of 10,000 CIFAR2 training labels on the first $K = 50$ eigenvectors of the kernel Gram matrices $\mathbf{\Phi}_t$ of a LeNet.

Figure 7: Alignment of the eigenfunctions of the NTK at initialization for a LeNet with the NTKs computed at the beginning (left) and at the end of 100 epochs of training (right) to predict the 500th eigenfunction at initialization.

the linearized models have a fixed kernel – and it hinges on a dynamical realignment of $\mathbf{\Theta}_t$ during training. That is, training a neural network rotates $\mathbf{\Theta}_t$ in a way that increases $\|\mathbf{\Phi}_t \mathbf{y}\|_2 / \|\mathbf{y}\|_2$.

Prior work has also observed a similar phenomenon, albeit in a more restricted experimental setup: These observations have been confirmed only in a few datasets, and required the use of minibatches to approximate the alignment of the NTK to track the evolution of a small portion of the eigenspace [22–24]. However, we now show that that the kernel rotation is prevalent across training setups, and that it can also be observed when training to solve other problems. In fact, a more fine-grained inspection reveals that the kernel rotation mostly happens in a single functional axis. It maximizes the alignment of $\mathbf{\Theta}_t$ with the target function $f$, i.e., $\alpha_t(f) = \|\mathbb{E}_{\boldsymbol{x}}[f(\boldsymbol{x})\nabla_{\boldsymbol{\theta}} f_{\boldsymbol{\theta}_t}(\boldsymbol{x})]\|_2^2$, but does not greatly affect the rest of the spectrum. Indeed, we can see how during training $\alpha_t$ grows significantly more for the target than for any other function.

This is clearly illustrated in Fig. 7, where we compare $\alpha_0$ and $\alpha_{100}$ for the first $1,000$ eigenfunctions of $\mathbf{\Theta}_0$, $\{\phi_j\}_{j=1}^{1,000}$, when training to predict and arbitrary eigenfunction[6] $\boldsymbol{x} \mapsto \text{sign}(\phi_{500}(\boldsymbol{x}))$. Strikingly, we can see that after training to predict $\phi_{500}$, $\alpha(\phi_{500})$ increases much more than $\alpha(\phi_j)$ for any other eigenfunction $\phi_j$. In fact, the relative alignment between all other eigenfunctions does not change much. Note, however, that the absolute values of all alignments have also grown, a phenomenon which is due to a general increase in the Jacobian norm $\mathbb{E}_{\boldsymbol{x}}\|\nabla_{\boldsymbol{\theta}} f_{\boldsymbol{\theta}_t}(\boldsymbol{x})\|_2$ during training.

These results show that there is great potential in using linear approximations to investigate important deep learning phenomena involving pretrained networks as in [10, 17]. Indeed, as $\alpha_t(f)$ is higher at the end of training, it can be expected that a linear expansion at this late stage will be able to capture the inductive bias needed to fine-tune on targets similar to $f$. The intuition behind this lies in the geometry of the NTK RKHS $\mathcal{H}$. Note that in $\mathcal{H}$, $\|\hat{f}_{\boldsymbol{\theta}}\|_{\mathbf{\Theta}} = \|\boldsymbol{\theta} - \boldsymbol{\theta}_0\|_2$ [3]; and recall that, as indicated by Lemma 1, target functions with small $\|f\|_{\mathbf{\Theta}}$ also have high $\alpha(f)$. This means that in $\mathcal{H}$ those tasks with a high $\alpha(f)$ are indeed represented by weights closer to the origin of the linearization, which thus makes the approximation error of the linearization smaller for these targets as observed in [21].

### 4.3 Kernel rotation improves speed of convergence, but can hurt generalization

The rotation of $\mathbf{\Theta}_t$ during training is an important mechanism that explains why the NTK dynamics can better capture the behavior of neural networks at the end of training. However, it is also fundamental in explaining the ability of neural networks to quickly fit the training data. Specifically, it is important to highlight the stark contrast, at a dynamical level, between linear models and neural networks. Indeed, we have consistently observed across our experiments that neural networks converge much faster to solutions with a near-zero training loss than their linear approximations.

We can explain the influence of the rotation of the NTK on this phenomenon through a simple experiment. In particular, we train three different models to predict another arbitrary eigenfunction $\boldsymbol{x} \mapsto \text{sign}(\phi_{400}(\boldsymbol{x}))$: i) a ResNet18, ii) its linearization around initialization, and iii) an unbiased linearization around the solution of the ResNet18 $\boldsymbol{\theta}^\star$, i.e., $\hat{f}_{\boldsymbol{\theta}}(\boldsymbol{x}) = (\boldsymbol{\theta} - \boldsymbol{\theta}_0)^\top \nabla_{\boldsymbol{\theta}} f_{\boldsymbol{\theta}^\star}(\boldsymbol{x})$.

Fig. 8 compares the dynamics of these models, revealing that the neural network indeed converges much faster than its linear approximation. We see, however, that the kernelized model constructed

---

[6]Recall that $\alpha_0(\phi_j) = \lambda_j$. Similar results for other networks and tasks can be found in the Appendix

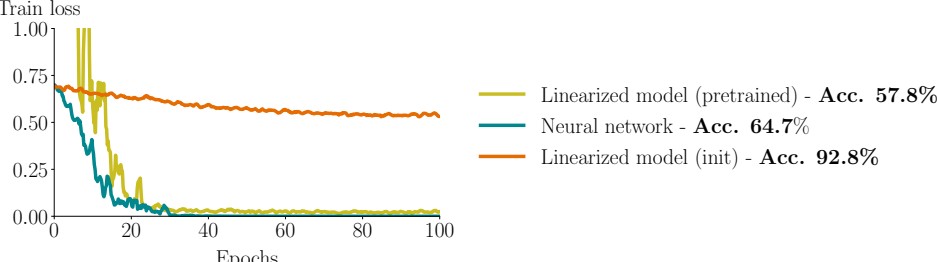

Figure 8: (Left) Evolution of training loss while learning $x \mapsto \mathrm{sign}(\phi_{400}(x))$ for a ResNet18 and two linearized models based on $\Theta_0$ and $\Theta_{100}$ of the ResNet18. (Right) Test accuracy on $x \mapsto \mathrm{sign}(\phi_{400}(x))$ for the three models.

using the pretrained NTK of the network has also a faster convergence. We conclude, therefore, that, since the difference between the two linear models only lies on the kernel they use, it is indeed the transformation of the kernel through the non-linear dynamics of the neural network that makes these models converge so quickly. Note, however, that the rapid adaptation of $\Theta_t$ to the training labels can have heavy toll in generalization, i.e., the model based on the pretrained kernel converges much faster than the randomly initialized one, but has a much lower test accuracy (comparable to the one of the neural network).

The dynamics of the kernel rotation are very fast, and we observe that the NTK overfits to the training labels in just a few iterations. Fig. 9 illustrates this process, where we see that the performance of the linearized models with kernels extracted after a few epochs of non-linear pretraining decays very rapidly. This observation is analogous to the one presented in [21], although in the opposite direction. Indeed, instead of showing that pretraining can greatly improve the performance of the linarized networks, Fig. 9 shows that when the training task does not abide by the inductive bias of the network, pretraining can rapidly degrade the linear network performance. On the other hand, on CIFAR2 (see Sec. 4.1) the kernel rotation does greatly improve test accuracy for some models.

The fact that the non-linear dynamics can both boost and hurt generalization highlights that the kernel rotation is subject to its own form of inductive bias. In this sense, we believe that explaining the non-trivial coupling between the kernel rotation, alignment, and training dynamics is an important avenue for future research, which will allow us to better understand deep networks.

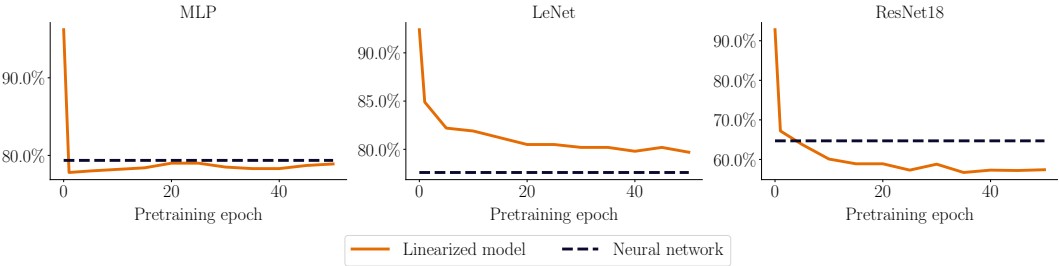

Figure 9: Performance of linearized networks with kernels extracted after different epochs of pretraining of a non-linear network learning $x \mapsto \mathrm{sign}(\phi_{400}(x))$. The dashed line represents the performance of fully non-linear training on the same task, and the value at epoch 0 corresponds to the linearized model at initialization.

## 5 Final remarks

Explaining generalization in deep learning [1] has been the subject of extensive research in recent years [34–38]. The NTK theory is part of this trend, and it has been used to prove convergence and generalization of very wide networks [2, 4–9]. Most of these efforts, however, still cannot explain the behavior of the models used in practice. Therefore, multiple authors have proposed to study neural networks empirically, with the aim to identify novel deep learning phenomena which can be later explained theoretically [21, 39–41].

In this work, we have followed a similar approach, and presented a systematic study comparing the behavior of neural networks and their linear approximations on different tasks. Previous studies had shown there exist tasks that neural networks can solve but kernels cannot [18–21]. Our work complements those results, and provides examples of tasks where kernels perform better than neural networks (see Sec.3.1). We see this result as an important milestone for deep learning theory, as it shifts the focus of our research from asking "why are non-linear networks better than kernel methods?" to "what is special about our standard tasks which makes neural networks adapt so well to them?". Moving forward, knowing which tasks a neural network can and cannot solve efficiently will be fundamental to explain their inductive bias [42].

Our findings complement the work in [21] where the authors also empirically compared neural networks and their linear approximations, but focused on a single training task. In this work, we have built from these observations and studied models on a more diverse set of problems. Doing so, we have shown that the alignment with the empirical NTK can rank the learning complexity of certain tasks for neural networks, despite being agnostic to the non-linear (dis)advantages. In this sense, we have revealed that important factors such as sample size, architecture, or target task can greatly influence the gap between kernelized models and neural networks.

Finally, our dynamical study of the kernel rotation complements the work of [22–24] and provides new important insights for future research. For example, the fact that the kernel rotates in a single axis, and that its tendency to overfit is accurately predicted by the NTK eigenfunctions can aid in the development of new models of training. Moreover, it opens the door to new algorithmic developments that could slow down the kernel rotation and potentially reduce overfitting on many tasks.

Overall, our work paves the way for new research avenues based on deep learning theory, while it also provides actionable insights to use the NTK approximation in many applications.

## Acknowledgements

We thank Emmanuel Abbé, Alessandro Favero, Apostolos Modas and Clément Vignac for their fruitful discussions and feedback.

## Funding disclosure

This work has been partially supported by Google via a Postdoctoral Fellowship and a GCP Research Credit Award.

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
