# What can linearized neural networks actually say about generalization?

## A  General training setup

As mentioned in the main text, all our models are trained using the same scheme which was selected without any hyperparameter tuning, besides ensuring a good performance on CIFAR2 for the neural networks. Namely, we train using stochastic gradient descent (SGD) to optimize a binary cross-entropy loss, with a decaying learning rate starting at $0.05$ and momentum set to $0.9$. Furthermore, we use a batch size of $128$ and train for a $100$ epochs. This is enough to obtain close-to-zero training losses for the neural networks, and converge to a stable test accuracy in the case of the linearized models[1]. In fact, in the experiments involving CIFAR2, we train all models for 200 epochs to allow further optimization of the linearized models. Nevertheless, even then, the neural networks perform significantly better than their linear approximations on this dataset.

In terms of models, all our experiments use the same three models: A multilayer perceptron (MLP) with two hidden layers of $100$ neurons each, the standard LeNet5 from [1], and the standard ResNet18 [2]. We used a single V100 GPU to train all models, resulting in training times which oscillated between $5$ minutes for the MLP, to around $40$ minutes for the ResNet18.

## B  NTK computation details

We now provide a few details regarding the computation of different quantities involving neural tangent kernels and linearized neural networks. In particular, in our experiments we make extensive use of the `neural_tangents` [3] library written in JAX [4], which provides utilities to compute the empirical NTK or construct linearized neural networks efficiently.

**Eigendecompositions**   As it is common in the kernel literature, in this work, we use the eigenvectors of the kernel Gram matrix to approximate the eigenfunctions of the NTK. Specifically, unless stated otherwise, in all our experiments we compute the Gram matrix of the NTK at initialization using the $60,000$ samples of the CIFAR10 dataset, which include both training and test samples. To that end we use the `empirical_kernel_fn` from `neural_tangents` which allows to compute this matrix using a batch implementation. Note that this operation is computationally intense, scaling quadratically with the number of samples, but also quadratically with the number of classes. For instance, in the single-output setting of our experiments[2], it takes up to 32 hours to compute the Gram matrix of the full CIFAR10 dataset for a ResNet18 using 4 V100 GPUs with 32Gb of RAM each. The computation for the LeNet and the MLP take only 20 and 3 minutes, respectively, due to their much smaller sizes.

**Linearization**   Using `neural_tangents`, training and evaluating a linear approximation of any neural network is trivial. In fact, the library already comes with a function `linearize` which allows to obtain a fully trainable model from any differentiable JAX function. Thus, in our experiments, we treat all linearized models as standard neural networks and use the same optimization code to train

---

[1]Note that the linearized models converge significantly slower than the neural networks.

[2]Note that the binary cross-entropy loss can be computed using a single output logit.

them. In the case of the ResNet18 network, which includes batch normalization layers [5], we fix the batch normalization parameters to their initialization values when performing the linearization. This effectively deactivates batch normalization for the linear approximations. Note that in [6] they also compared neural networks with batch normalization to their linear approximations without it.

**Alignment**   Obtaining the full eigendecomposition of the NTK Gram matrix is computationally intense and only provides an approximation of the true eigenfunctions. However, in some cases it is possible to compute some of the spectral properties of the NTK in a more direct way, thus circumventing the need to compute the Gram matrix. This is for example the case for the target alignment $\alpha(f)$, which using the formula in Lemma 1 can be computed directly using a weighted average of the Jacobian. This is precisely the way in which we computed $\alpha(\phi_j)$ for different eigenfunctions $\{\phi_j\}_{j=1}^{1000}$ in Sec. 4.2.

**Binarization**   In most of our experiments we do not work directly with the eigenfunctions of the NTK, but rather with their binarized versions, i.e., $\text{sign}(\phi_j(\boldsymbol{x}))$. Nevertheless, we would like to highlight that this transformation has little effect on the direction of the targets: In such high-dimensional setting, the binarized and normal eigenvectors have an inner product of approximately 0.78, while the average inner product between random vectors is of the order of $10^{-4}$.

## C  Neural anisotropy experiments

### C.1  Dataset construction

We used the same linearly separable dataset construction proposed in [7] to test the NADs computed using Theorem 1. Specifically, for every NAD $\boldsymbol{u}$ we tested, we sampled $10,000$ training samples from $(\boldsymbol{x}, y) \sim \mathcal{D}(\boldsymbol{u})$ where

$$\boldsymbol{x} = \epsilon\, y\, \boldsymbol{u} + \boldsymbol{w} \quad \text{with} \quad \boldsymbol{w} \sim \mathcal{N}(\boldsymbol{0}, \sigma(\boldsymbol{I} - \boldsymbol{u}\boldsymbol{u}^\top)) \quad \text{and} \quad y \sim \mathcal{U}\{-1, 1\}. \tag{1}$$

As in [7], we used a value of $\epsilon = 1$ and $\sigma = 1$, and tested on another $10,000$ test samples from the same dataset.

### C.2  NAD computation

We provide a few visual examples of the NADs computed for the LeNet and ResNet18 networks using the singular value decomposition (SVD) of the mixed second derivative. Specifically, in our preliminary experiments we did not find any significant difference in the NADs computed using the average over the data or at the origin, and hence opted to perform the SVD over $\nabla^2_{\boldsymbol{x},\boldsymbol{\theta}} f_{\boldsymbol{\theta}_0}(0)$ instead of $\mathbb{E}_{\boldsymbol{x}}[\nabla^2_{\boldsymbol{x},\boldsymbol{\theta}} f_{\boldsymbol{\theta}_0}(0)]$. This was also done in [7], and can be seen as a first order approximation of the expectation. Furthermore, to avoid the non-differentiability problems of the ReLU activations described in [7] we used GeLU activations [8] in these experiments. A few examples of the NADs computed using this procedure can be found in Figure 1.

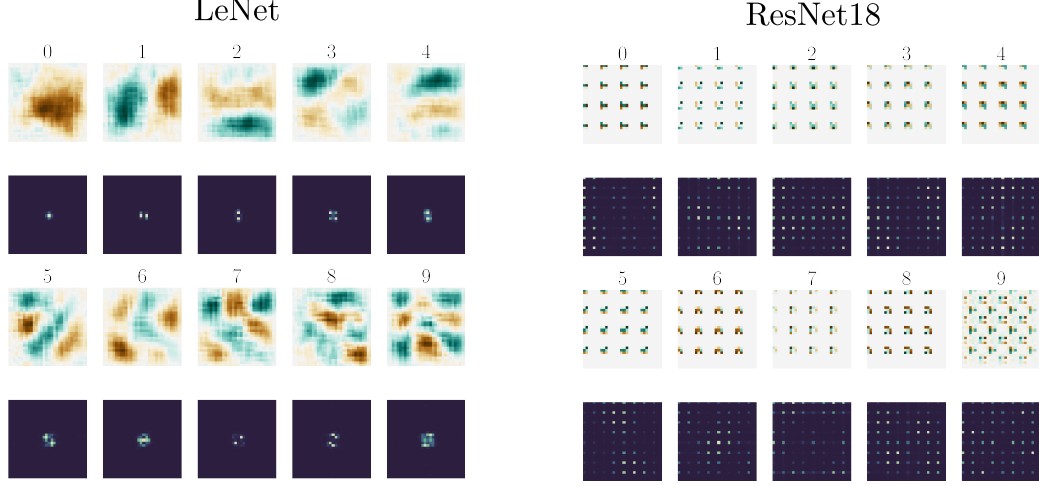

Figure 1: First 10 NADs computed using Theorem 1 of a randomly initialized LeNet and a ResNet18. As in [7], we show both the spatial domain (light images) and the magnitude of the Fourier domain (dark images) for each NAD.

# D    Differed proofs

## D.1    Proof of Lemma 1

We give here the proof of Lemma 1 which gives a tractable bound on the kernel norm of a function $f \in \mathcal{H}$ based on its alignment with the kernel. We restate the theorem to ease readability.

**Lemma.** *Let $\alpha(f) = \mathbb{E}_{\boldsymbol{x},\boldsymbol{x}'\sim\mathcal{D}} \left[ f(\boldsymbol{x})\Theta(\boldsymbol{x},\boldsymbol{x}')f(\boldsymbol{x}') \right]$ denote the alignment of the target $f \in \mathcal{H}$ with the kernel $\Theta$. Then $\|f\|_{\Theta}^2 \geq \|f\|_2^4/\alpha(f)$. Moreover, for the NTK, $\alpha(f) = \|\mathbb{E}_{\boldsymbol{x}} \left[ f(\boldsymbol{x})\nabla_{\boldsymbol{\theta}} f_{\boldsymbol{\theta}_0}(\boldsymbol{x}) \right]\|_2^2$.*

*Proof.* Given the Mercer's decomposition of $\Theta$, i.e., $\Theta(\boldsymbol{x},\boldsymbol{x}') = \sum_{j=1}^{\infty} \lambda_j \phi_j(\boldsymbol{x})\phi_j(\boldsymbol{x}')$, the alignment of $f$ with $\Theta$ can also be written

$$\alpha(f) = \mathbb{E}_{\boldsymbol{x},\boldsymbol{x}'\sim\mathcal{D}} \left[ \sum_{j=1}^{\infty} \lambda_j \phi_j(\boldsymbol{x})\phi_j(\boldsymbol{x}')f(\boldsymbol{x})f(\boldsymbol{x}') \right] = \sum_{j=1}^{\infty} \lambda_j \left( \mathbb{E}_{\boldsymbol{x}\sim\mathcal{D}} \left[ \phi_j(\boldsymbol{x})f(\boldsymbol{x}) \right] \right)^2 . \tag{2}$$

Now, recall that for a positive definite kernel, the kernel norm of a function admits the expression

$$\|f\|_{\Theta}^2 = \sum_{j=1}^{\infty} \frac{1}{\lambda_j} \left( \mathbb{E}_{\boldsymbol{x}\sim\mathcal{D}} \left[ \phi_j(\boldsymbol{x})f(\boldsymbol{x}) \right] \right)^2 . \tag{3}$$

The two quantities can be related using Cauchy-Schwarz to obtain

$$\sqrt{\sum_{j=1}^{\infty} \frac{1}{\lambda_j} \left( \mathbb{E}_{\boldsymbol{x}\sim\mathcal{D}} \left[ \phi_j(\boldsymbol{x})f(\boldsymbol{x}) \right] \right)^2} \sqrt{\sum_{j=1}^{\infty} \lambda_j \left( \mathbb{E}_{\boldsymbol{x}\sim\mathcal{D}} \left[ \phi_j(\boldsymbol{x})f(\boldsymbol{x}) \right] \right)^2} \geq \sum_{j=1}^{\infty} \left( \mathbb{E}_{\boldsymbol{x}\sim\mathcal{D}} \left[ \phi_j(\boldsymbol{x})f(\boldsymbol{x}) \right] \right)^2$$

$$\tag{4}$$

$$\|f\|_{\Theta} \sqrt{\alpha(f)} \geq \|f\|_2^2. \tag{5}$$

On the other hand, in the case of the NTK

$$\begin{aligned}
\alpha(f) &= \mathbb{E}_{\boldsymbol{x},\boldsymbol{x}'\sim\mathcal{D}} \left[ f(\boldsymbol{x})f(\boldsymbol{x}')\nabla_{\boldsymbol{\theta}}^{\top} f_{\boldsymbol{\theta}_0}(\boldsymbol{x})\nabla_{\boldsymbol{\theta}} f_{\boldsymbol{\theta}_0}(\boldsymbol{x}') \right] \\
&= \mathbb{E}_{\boldsymbol{x}\sim\mathcal{D}} \left[ f(\boldsymbol{x})\nabla_{\boldsymbol{\theta}}^{\top} f_{\boldsymbol{\theta}_0}(\boldsymbol{x}) \right] \mathbb{E}_{\boldsymbol{x}'\sim\mathcal{D}} \left[ f(\boldsymbol{x}')\nabla_{\boldsymbol{\theta}} f_{\boldsymbol{\theta}_0}(\boldsymbol{x}') \right] = \left\| \mathbb{E}_{\boldsymbol{x}} \left[ f(\boldsymbol{x})\nabla_{\boldsymbol{\theta}} f_{\boldsymbol{\theta}_0}(\boldsymbol{x}) \right] \right\|_2^2 . \tag{6}
\end{aligned}$$

$\square$

## D.2    Proof of Theorem 1

We give here the proof of Theorem 1 which gives a closed form expression of the alignment of a linear predictor with the NTK. We restate the theorem to ease readability.

**Theorem.** *Let $\boldsymbol{u} \in \mathbb{S}^{d-1}$ be a unitary vector that parameterizes a linear predictor $g_{\boldsymbol{u}}(\boldsymbol{x}) = \boldsymbol{u}^{\top}\boldsymbol{x}$, and let $\boldsymbol{x} \sim \mathcal{N}(\boldsymbol{0}, \boldsymbol{I})$. The alignment of $g_{\boldsymbol{u}}$ with $\Theta$ is given by*

$$\alpha(g_{\boldsymbol{u}}) = \left\| \mathbb{E}_{\boldsymbol{x}} \left[ \nabla_{\boldsymbol{x},\boldsymbol{\theta}}^2 f_{\boldsymbol{\theta}_0}(\boldsymbol{x}) \right] \boldsymbol{u} \right\|_2^2 , \tag{7}$$

*where $\nabla_{\boldsymbol{x},\boldsymbol{\theta}}^2 f_{\boldsymbol{\theta}_0}(\boldsymbol{x}) \in \mathbb{R}^{n \times d}$ denotes the derivative of $f_{\boldsymbol{\theta}}$ with respect to the weights and the input.*

*Proof.* Plugging the definition of a linear predictor on the expression of the alignment for the NTK (see Lemma 1) we get

$$\alpha(f) = \left\| \mathbb{E}_{\boldsymbol{x}} \left[ \nabla_{\boldsymbol{\theta}} f_{\boldsymbol{\theta}_0}(\boldsymbol{x})\boldsymbol{x}^{\top}\boldsymbol{u} \right] \right\|_2^2 , \tag{8}$$

which using Stein's lemma becomes

$$\alpha(f) = \left\| \mathbb{E}_{\boldsymbol{x}} \left[ \nabla_{\boldsymbol{\theta},\boldsymbol{x}}^2 f_{\boldsymbol{\theta}_0}(\boldsymbol{x})\boldsymbol{u} \right] \right\|_2^2 . \tag{9}$$

$\square$

# E Additional results

We now provide the results of some experiments which complement the findings in the main text.

## E.1 Learning NTK eigenfunctions

In Section 3.2, we saw that neural networks and their linear approximations share the way in which they rank the complexity of learning different NTK eigenfunctions. However, this experiments were performed using a single training setup, and a single data distribution. For this reason, we now provide two complementary sets of experiments which highlight the generality of the previous results.

On the one hand, we repeated the experiment in Section 3.2. using a different training strategy, replacing SGD with the popular Adam [9] optimization algorithm. As we can see in Figure 2 the main findings of Section 3.2. also transfer to this different training setup. In particular, we see that the performance of all models progressively decays with increasing eigenfunction index, and that the linearized models have a clear *linear advantage* over the non-linear neural networks.

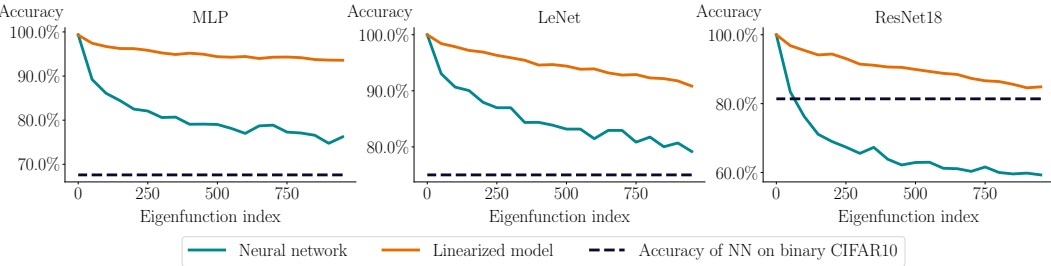

Figure 2: Validation accuracy of different neural network architectures and their linearizations when trained on binarized eigenfunctions of the NTK at initialization, i.e., $x \mapsto \mathrm{sign}(\phi_j(x))$ using Adam. As a baseline, we also provide the accuracies on CIFAR2.

On the other hand, we also repeated the same experiments changing the underlying data distribution, and instead of using the CIFAR10 samples, we used the MNIST [1] digits. The results in Figure 3 show again the same tendency.[3] However, we now see, that for the LeNet, the accuracy curves of the neural network and the linearized model cross around $\phi_{500}$, highlighting that the existence of a *linear* or *non-linear* advantage greatly depends on the target task.

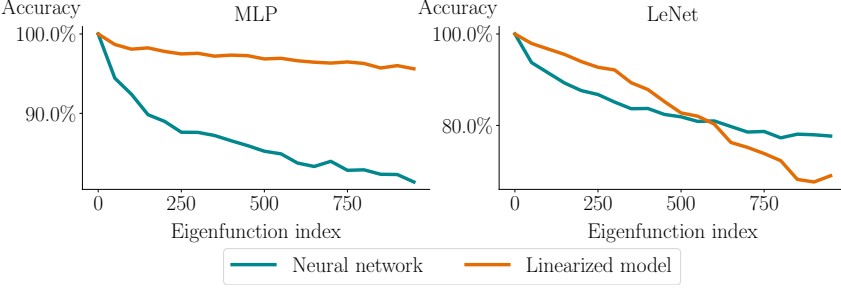

Figure 3: Validation accuracy of different neural network architectures and their linearizations when trained on binarized eigenfunctions of the NTK at initialization, i.e., $x \mapsto \mathrm{sign}(\phi_j(x))$ computed over MNIST data.

---

[3]Note that the MNIST dataset has more samples than CIFAR10 (i.e., $70,000$ samples), and hence, due to the quadratic complexity of the Gram matrix computation and budgetary reasons, we decided to not perform this experiment on ResNet18.

### E.1.1 Convergence speed of other networks

We also provide the collection of training metrics of the MLP (see Figure 4) and the LeNet (see Figure 5) trained on the different eigenfunctions of $\boldsymbol{\Theta}_0$. Again, as was the case for the ResNet18, we see that training is "harder" for the eigenfunctions corresponding to the smaller eigenvalues, as the time to reach a low training loss, and the distance to the weight initialization grow with eigenfunction index.

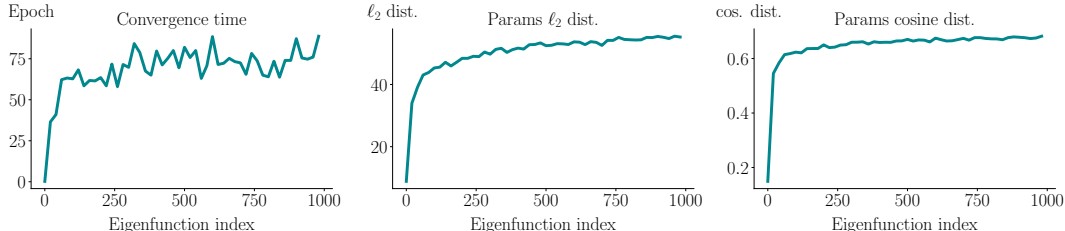

Figure 4: Correlation of different training metrics with the index of the eigenfunction the network is trained on. Plots show the number of training iterations taken by the network to achieve a 0.01 training loss, and the $\ell_2$ and cosine distances between initialization and final parameters for a MLP trained on the binarized eigenfunctions of the NTK at initialization.

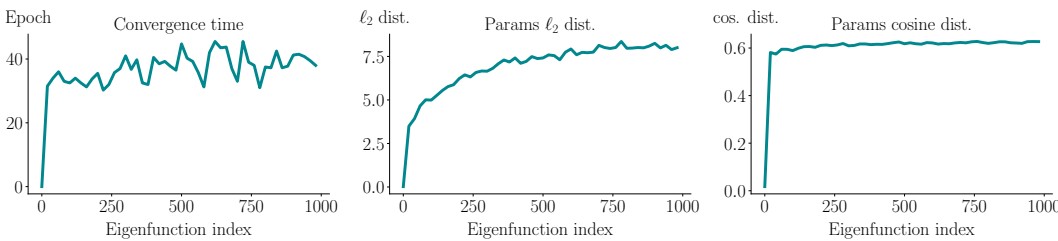

Figure 5: Correlation of different training metrics with the index of the eigenfunction the network is trained on. Plots show the number of training iterations taken by the network to achieve a 0.01 training loss, and the $\ell_2$ and cosine distances between initialization and final parameters for a LeNet trained on the binarized eigenfunctions of the NTK at initialization.

### E.2 Energy concentration of CIFAR2 labels

We have seen that training a network on a task greatly increases the energy concentration of the training labels with the final kernel. In the main text, however, we only provided the results for the evolution of the energy concentration, i.e., $\|\boldsymbol{\Phi}_t \boldsymbol{y}\|_2 / \|\boldsymbol{y}\|_2$, for the LeNet trained on CIFAR2. Table 1 shows the complete results for the other networks, clearly showing an increase in the energy concentration at the end of training.

Table 1: Energy concentration on top $K = 50$ eigenvectors of the NTK Gram matrices at initialization $\boldsymbol{\Theta}_0$ and in the last epoch of training $\boldsymbol{\Theta}_{200}$ computed on $12,000$ training samples. We also provide the test accuracy on CIFAR2 for comparison.

|  | MLP | LeNet | ResNet18 |
| --- | --- | --- | --- |
| Energy conc. (init) | 26.0% | 25.8% | 22.7% |
| Energy conc. (end) | 63.6% | 83.0% | 96.7% |
| Test accuracy | 67.6% | 75.0% | 81.4% |

### E.3 Increase of alignment when learning NTK eigenfunction

Finally, we provide the final alignment plots of the networks trained to predict different eigenfunctions than the one showed in the main text. As we can see in Figure 6 and Figure 7 the relative alignment

of the training eigenfunction spikes at the end of training, demonstrating the single-axis rotation of the empirical NTK during training.

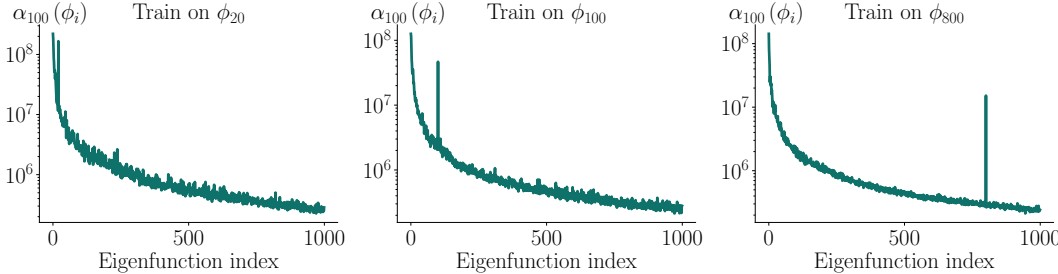

Figure 6: Alignment of the eigenfunctions of the NTK of a randomly initialized MLP with the NTK computed after training to predict different initialization eigenfunctions.

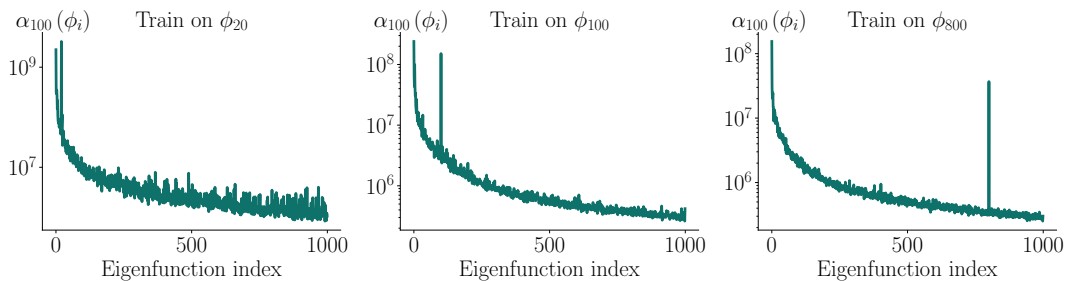

Figure 7: Alignment of the eigenfunctions of the NTK of a randomly initialized LeNet with the NTK computed after training to predict different initialization eigenfunctions.