# OpenReview forum: "What can linearized neural networks actually say about generalization?"
_NeurIPS.cc/2021/Conference — NeurIPS 2021 Poster_

### Official Review · Reviewer_rfX8 · 2021-07-08

**Rating:** 5
**Confidence:** 5

**Summary:**

This paper studies the training dynamics of both neural networks and their linear approximations, and show that the finite-width NTK aligns to the target function during training, which in turn induces rapid convergence to a particular solution.

**Limitations And Societal Impact:**

yes

**Main Review:**

While this submission is well-written and covers an interesting avenue of research, most of its content has previously been published in a AISTAT2021 paper: Baratin et al., namely:

 - the dynamical bias of linear models towards functions that are aligned with the kernel (which is a standard result)
 - the increase in alignment of the NTK to the target during training of a neural network
 - the consequence of this increase in alignment on (linear) generalization guarantees

Therefore, depending on the NeurIPS policy regarding prior work, it looks to me that this work cannot be considered an original contribution. I am putting a low score only for that reason.

A comment regarding the experiment in section 4.1/fig 5: As you increase the training set size while keeping the minibatch size fixed, you also increase the number of minibatches in an epoch, thus there are more training iterations for the same number of epochs. Given that, I am not so surprised by the increasing l2 distance to initialization. Do these different behaviours between linear and non-linear dynamics occur because of the increasing number of examples, or because of the increasing number of training iterations?


Reference:
Baratin, George, Laurent, Hjelm, Lajoie, Vincent, and Lacoste-Julien, Implicit regularization via Neural Feature Alignment, AISTATS2021 https://arxiv.org/abs/2008.00938

**Time Spent Reviewing:**

1.5

---

> ### Author Response · Authors · 2021-08-10
> **Reply to Reviewer rfX8**
>
> Thank you very much for having taken the time to review our paper. We sincerely appreciate that you believe our article is well-written.
>
> We agree that it is very unfortunate that we missed Baratin et al. in our previous literature search, and we will make sure to properly reference its contributions in the final version of our manuscript. However, ***we strongly dispute the view that our work is not novel***, and it is not true that our work just repeats previous findings.
>
> On the one hand, as correctly pointed out by [Reviewer 4dYZ](https://openreview.net/forum?id=KLS346_Asf&noteId=3aQDVU6pIPp), our work deals with a diverse set of topics, only a small fraction of it intersects with Baratin et al., and it does so to provide new important insights, which are:
> 1. The kernel adaptation can sometimes hurt generalization
> 2. The kernel rotation happens mostly in a single axis
>
> We firmly believe these two novel observations are of extreme relevance to the community and, to the best of our knowledge, our work has been the first to discuss them.
>
> On the other hand, a major part of our paper reveals that linear and non-linear models agree on the way they rank learning complexity, it shows the non-linear advantage is not ubiquitous among learning tasks, and it sheds new light on the directional inductive bias of neural networks (i.e., neural anisotropy directions). This part is completely orthogonal to Baratin et al. and gives a different perspective on the main message of our paper “Neural networks are not always superior to their linear approximations” which goes well beyond Baratin et al.
>
> For all these reasons, we urge the reviewer to please reassess the originality of our work, and to please provide a more holistic evaluation of our paper.
>
> > A comment regarding the experiment in section 4.1/fig 5: As you increase the training set size while keeping the minibatch size fixed, you also increase the number of minibatches in an epoch, thus there are more training iterations for the same number of epochs. Given that, I am not so surprised by the increasing l2 distance to initialization. Do these different behaviours between linear and non-linear dynamics occur because of the increasing number of examples, or because of the increasing number of training iterations?
>
> If instead of fixing number of epochs, one fixes the number of training iterations such that all networks achieve a at least a given small training loss, one also sees a clear increasing trend in $\ell_2$ distance to initialization both for the linear and non-linear networks. That is, *it is easier for the networks to find solutions close to initialization for smaller training set sizes even when training for exactly the same number of iterations*.
>
> However, it is important to note that linear and non-linear models achieve their maximum test accuracy in roughly the same number of epochs regardless of the training set size. This contrasts with the dynamics of the training loss, for which the linear models take significantly more iterations to perfectly fit the training data than non-linear ones. In this sense, the results of Fig. 5 present a snapshot taken when both networks approximately achieve their maximum generalization performance.

---

> > ### Comment · Reviewer_rfX8 · 2021-08-30
> > **Judging on the contributions other than the alignment of the empirical NTK**
> >
> > Thanks for your answer, and sorry for a harsh initial rating for a work for which I would be very enthusiastic had the most important claimed contribution (the alignment of the empirical NTK to the target) not been already described in other papers.
> >
> > I agree that other than section 4.2, you contribute some other interesting observations:
> > 1. relative complexity of tasks using deep learning can be predicted using the empirical NTK at init
> > 2. toy tasks where NNs perform worse than linear models
> > 3. the alignment can hurt generalization
> > I am here summarizing your lines 61-71, I removed the part regarding NTK alignment.
> >
> > Claim #1 and #2 are well discussed, but I feel that since this is a mostly empirical observation, claim #3 should be strengthened by a more diverse set of experiments (e.g. how does accuracy of the linear model using the empirical NTK vary as you extract it at several different iterations during training).
> >
> > Judging on this list of contributions, I updated my rating. I still feel like this paper should be improved to be published at NeurIPS.
> >
> > Note: In addition to the papers mentioned by rev 4dYZ, Paccolat et al (already present as [26] in your paper) also note the increasing alignment of the empirical NTK during training (named "compression" in their work).

---

> > > ### Author Response · Authors · 2021-08-31
> > > **Reply to new concerns**
> > >
> > > Thank you very much for your comment. We sincerely appreciate the clear change of mind, and positively welcome your new feedback. Considering the limited time left for the discussion, we will try our best to address your new concerns.
> > >
> > > In particular, we firmly contend your view on the need of more experimentation to support claim #3. This claim is supported by the experiments in Fig. 2 and Fig. 8 in the main text, and Fig. 2 and Fig. 3 in the supplementary material. Specifically, we have demonstrated the existence of a non-linear disadvantage on:
> > >
> > > -   Three different architectures (Fig.2 and E.1. in supp material)
> > > -   Two supporting datasets (see E.1. in supp. material)
> > > -   Using different optimizers (see E.1. in supp. material)
> > > -   50 tasks per network (i.e., different eigenfunctions)
> > >
> > > Furthermore, we have performed our experiments using the full eigendecomposition of the Gram matrix of the datasets, a very computationally intense procedure which, to the best of our knowledge, no other work had done before (previous work had always worked with batched approximations of the Gram matrix).
> > >
> > > We would also like to emphasize that many influential papers in deep learning theory also exclusively rely on empirical results to validate their findings; sometimes even, just experimenting on one network or a single dataset. This did not prevent those works from being published at previous editions of NeurIPS or similar venues, paving the way for new theoretical discoveries later on.
> > >
> > > Finally, we are happy to share that we have managed to run the only additional experiment you demanded, and would like to share its results (which we would be open to include in the final version of our paper). Specifically, we have replicated the same setup as in Fig. 8, but linearizing the network at different stages of training instead of just at the end. The following table shows the final test accuracy achieved by the linearized LeNet and ResNet18 at different stages of training:
> > >
> > > |          | Init  | Ep. 1 | Ep. 5 | Ep. 10 | Ep. 15 | Ep. 20 | Ep. 25 | Ep. 30 | Ep. 35 | Ep. 40 | Ep. 45 | Ep. 50 | Ep. 100 | Non-linear |
> > > |----------|-------|-------|-------|--------|--------|--------|--------|--------|--------|--------|--------|--------|---------|------------|
> > > | LeNet    | 92.4% | 84.9% | 82.2% | 81.9%  | 81.2%  | 80.5%  | 80.5%  | 80.2%  | 80.2%  | 79.8%  | 80.2%  | 79.7%  | 77.3%   | 77.6%      |
> > > | ResNet18 | 92.8% | 67.2% | 63.8% | 60.1%  | 58.9%  | 58.9%  | 57.3%  | 58.8%  | 56.7%  | 57.3%  | 57.2%  | 57.4%  | 57.8%   | 64.7%      |
> > >
> > > Indeed, as described in Fig. 6, the kernel rotation is an extremely fast process, which happens in just a few steps of training. In the setup of this experiment, this translates into a fast drop in final test accuracy of the linearized networks, which gets worse with more pretraining. Note that this experiment is analogous to the one presented in [21]. However, this time, instead of showing that pretraining can greatly improve the performance of the linarized networks when the training task falls within the inductive bias of the kernel rotation, we show that when the training task does not abide by the inductive bias of the network, pretraining can rapidly degrade the linear network performance.
> > >
> > > All in all, we honestly believe that our answer fully addresses all your new comments: We have clarified the thoroughness of our experimental setup, and provided the experiments you demanded. In this sense, we actively urge this reviewer to increase their score and support the acceptance of our paper.

---

### Official Review · Reviewer_NLbM · 2021-07-14

**Rating:** 5
**Confidence:** 2

**Summary:**

Disclaimer: I'm only superficially familiar with the related literature, so this is rather low-confidence review.

The paper investigates whether the study of linearized neural networks via the neural tangent kernel is insightful for understanding generalization in deep learning. It argues that this is indeed the case based primarily on empirical evidence. Overall I am not fully convinced by the significance of the results in this paper, so that I am leaning towards a reject, but I would expect the other reviews to be more informative.

**Main Review:**

Overall, I think this is a polished and well-written paper. Considering the growing body of recent works on the NTK the topic of the submission is clearly of interest to the community. In particular the paper attempts to address recent concerns over the NTK not being informative regarding the generalization capabilities of finite-width neural networks.

To this end, the paper presents the following results:
- Based on labels generated from the eigenfunctions of the NTK, both linearized and finite-width NNs decrease in performance with increasing complexity of the learning taks. Further it observes that similarly on the recently proposed neural anisotropy directions performance decreases concurrently with increasing task complexity. These results complement previous works which have used the NTK to predict task complexities and gives them a slightly stronger theoretical foundation. However, even though the experiments are based on carefully chosen synthetic labels, the support is primarily empirical, which seems somewhat incremental and not nearly as strong as any theoretical guarantees.
- Next, the paper finds that neural networks primarily outperform their linearized counterparts in the large data regimes and argues that this is due to the distance of the optimum from initialization, which impacts the accuracy of the linearization approximation. It also notes that the alignment of the labels with the top eigenfunctions of the NTK increases through training.

While the results in this paper seem to fill some gaps in the literature, they feel somewhat incremental and mostly appear to add to previous observations rather than making novel ones of their own. Given this incremental nature I find that the predominantly empirical analysis lacks significance. If the authors think that this understates their contribution, I would encourage them to clarify what avenues of future research their work opens up specifically. The discussion at the end remains rather vague on this and (to my taste) focuses too much on summarizing the paper.

**Time Spent Reviewing:**

2.5

---

> ### Author Response · Authors · 2021-08-10
> **Reply to Reviewer NlbM**
>
> Thank you very much for the honest review and for giving us the opportunity to strengthen the importance of our work, and expand on what we think are the most promising research avenues that we have opened:
>
> 1. **Inductive bias of kernel rotation:** To the best of our knowledge, our work has been the first to show that neural networks are not uniformly superior to their linear approximations in all tasks. Besides, we have shown that one of the reasons for this is an extremely rapid adaptation of the kernel to the training data. As mentioned before, this is an important milestone in deep learning theory, as it shifts the focus of our research from asking "why are non-linear networks better than kernel methods?" to "what is special about our standard tasks, which makes neural networks adapt so well to them?". In this sense, our dynamical study of the kernel rotation which shows a single axis of rotation of the kernel, and a quantitative difference in the speed of adaptation depending on the initial eigenfunctions of the kernel, can provide important hints for future theoretical and algorithmic developments. One possible idea would be to try to slow down the kernel rotation through regularization so it does not adapt so quickly to the target task and hence reduce overfitting.
>
> 2. **Linearized models to study pre-trained networks:** Our work has shown that linear and non-linear models agree better in the predictions whenever the target task is highly aligned with the initial kernel. Hence, since pre-trained networks tend to be more aligned with the target task, our work opens the door to new research to understand the features learned by pre-trained models using kernel methods. This is an important topic of interest for transfer learning, which has already started to attract some attention [10].
>
> 3. **Directional inductive bias of neural networks:** In this paper, we have shown that one can successfully used the alignment with the NTK at initialization as a good proxy to compute neural anisotropy directions (NADs), a surprising phenomenon which seems to be pervasive across top-ranking models, but for which little theoretical explanations exist. The fact that the NTK of SOTA network presents such strong directional bias is remarkable on its own, as it reveals important structure of the underlying neural networks. Recent theoretical studies [a], have found that the rotational invariance assumption of standard kernel theory results might be too restrictive to explain generalization in many settings, and as such, showing that the kernels of neural networks have a strong rotational variance, clearly strengthens the link between the study of these kernels and deep learning theory. This connection can be used, for example, to derive new generalization bounds on deep neural networks which take into account their rotational variance.
>
> We hope these explanations have helped clarify not only the most important insights of our paper, but also how they can pave the way for new research avenues such as the regularization of the kernel rotation, or the derivation of new generalization bounds.
>
> [a] https://arxiv.org/abs/2104.04244

---

> > ### Comment · Reviewer_NLbM · 2021-08-24
> > **Re: Reply**
> >
> > Thank you for clarifying the significance of your work. Having also read the other reviews, I will keep my rating for now due to my low confidence, but remain open to increasing the score if this creates a consensus between the reviewers.

---

> > > ### Author Response · Authors · 2021-08-31
> > > **Update score to help reach an agreement**
> > >
> > > Thank you for your answer, and for being open to increasing your score. Specifically, considering that [Reviewer rfX8](https://openreview.net/forum?id=KLS346_Asf&noteId=SVS-fcIPgxW) has recognized they were harsh in their previous review, significantly increased their score, and that [Reviewer 4dYZ](https://openreview.net/forum?id=KLS346_Asf&noteId=JpdpV3IliCB) is clearly supporting the acceptance of our work, we would like to encourage you to reconsider your previous assessment and politely ask you to increase your score. In this sense, we remain open to clarify any outstanding doubt or concern you may have, and sincerely thank you for your work on this review.

---

### Official Review · Reviewer_4dYZ · 2021-07-16

**Rating:** 7
**Confidence:** 3

**Summary:**

The submitted paper studies the differences in the behavior of typical neural networks and the predictions made by the Neural Tangent Kernel (NTK) framework by showing that the NTK evolves and adapts to the current task during training. The experiments show that the predictions of NTK framework correctly rank tasks in terms of hardness but over- and under-estimate their hardness in different cases. The paper then studies the role of sample size and the directions where NTK evolves.

**Limitations And Societal Impact:**

The authors address limitations and societal impact.

**Main Review:**

The main finding of the paper, that NTK evolves through the training to align its top eigenvectors to the task, has already been shown in recent work. There are still new and interesting results about how NTK evolves and its potential for overfitting and I'm leaning towards acceptance.

Adaptation of NTK to the task has been recently shown by [1] (also see [2] for an empirical study). Those papers also talk about benefits of alignment for convergence and generalization. I have not read those two papers in depth and I'm happy to discuss if the authors believe the submitted paper is showing a different phenomenon.

The most interesting result to me was Figure 8 that shows NTK alignment can hurt generalization. While some previous works describe NTK alignment as a sort of implicit regularizer and also the generalization bound in Eq (2) suggests that alignment helps generalization, Figure 8 shows that the story is more complicated and allowing the network to evolve its kernel can hurt generalization. Do the authors have an explanation for this apparent paradox? I suspect that in small data regime the NTK itself overfits to the training data so that its alignment increases on train data but stays constant (or maybe even decreases) on test data. A plot like Figure 6 but evaluated on both train and test data and in a setting where NTK evolution hurts generalization can clear this up.

Another interesting finding is Figure 7 (and the replications in the appendix) where NTK alignment happens mostly in a single direction, although I'm not sure how this would extend to more general cases. For example, if the target function was aligned with two eigenvectors of the initial NTK, would the evolution be in those two directions or still in the direction of one eigenvector?

The experiments highlight the role of alignment on convergence while the theory in the introduction only motivates alignment with a generalization bound. Bringing an alignment-dependent convergence analysis beside the generalization bound can help with motivating alignment. An example is Theorem 4.1 in [3] (that paper is about ranking of tasks in terms of hardness and does not study evolution of NTK).

Minor comments:

In some experiments like section 3.1 the eigenvector has to be binarized to be used as training labels. Any idea how much this binarization changes the direction of this vector?

It looks to me that the curve in Figure 7 (left) becomes equal to lambda^2. Mentioning this in the text can help with understanding the plot. For the curve on the right, it is good to mention that phi_i still refers to eigenvectors of the initial NTK if this is the case.

The choice of K=50 for Figure 6 looks arbitrary and the result might be different for another K. Can one use the scalar measure in Lemma 1 to get rid of K?


[1] https://arxiv.org/abs/2008.00938

[2] https://arxiv.org/abs/1910.08720

[3] https://arxiv.org/abs/1901.08584

Update: increased the score as the comment about related work is addressed.

**Time Spent Reviewing:**

10

---

> ### Author Response · Authors · 2021-08-10
> **Reply to Reviewer 4dYZ**
>
> Thank you very much for the thorough review and abundant constructive feedback. We will try to address all your questions below.
>
> **Novelty of contribution:** Indeed, [1] and [2] are very relevant references for the latter half of our paper, and quite unfortunately, we missed them in our literature search. We will, therefore, make sure to properly reference their contributions in the final version of our manuscript and acknowledge these works that had previously observed the kernel rotation.
>
> Still, ***we firmly believe that our work adds new important insights wrt [1,2]***, which are fundamental to our understanding of deep learning. Specifically, a recurrent theme in the recent deep learning theory literature [1,2 but also 18-21 in-text] has been to provide examples in which neural networks outperform kernel methods, to the lengths that it has been strongly conjectured that non-linear models are always superior to their linear approximations. Our work, however, shows otherwise, as it provides several clear counterexamples of this premise. This is, in our honest opinion, an important milestone in our understanding of deep learning, as it shifts the focus of our research from "why are non-linear networks better than kernel methods?" to "what is special about our standard tasks, which makes neural networks adapt so well to them?".
>
> Similarly, our insights on the dynamics of the kernel rotation (i.e., it mostly happens in a single axis) are also complementary to [1,2].
>
> Finally, we would like to emphasize that besides the results on kernel rotation, a large part of our paper actually presents a novel and systematic study showing that linear and non-linear networks agree on the way they rank the hardness of learning tasks. An observation that strengthens prior work which uses linearized models as proxies for neural networks, and which enables us to shed new light on the existence of the neural anisotropy directions.
>
> All in all, based on the diversity and strength of these contributions, we believe our work is worthy of a NeurIPS publication, which we hope can pave the way to a better understanding of deep learning and inspire future research.
>
> **Why can kernel rotation hurt generalization?:** Our experiments indicate that the dynamics of the kernel rotation have a very strong implicit bias which does not fully match the NTK one. In this sense, when the network is asked to fit some "late" eigenfunction, the kernel rotation seems to prevent the network from identifying the generalizing signal in the training samples and ends up overfitting much faster to the training data.
>
> Surprisingly, however, when we performed a similar evaluation as the one proposed by the reviewer, analyzing the evolution of the alignment for train and test data separately, we never observed a decrease in alignment of the test samples. One can notice, however, that the alignment of the test samples is significantly lower (approximately by an order of magnitude) than in the training samples. Furthermore, we also observed (see Appendix Fig. 6 and 7) that the maximum alignment reached was much higher for the first eigenfunctions (easy to generalize) than for the last ones (harder to generalize).
>
> **Single direction of rotation:** We have repeated our experiments with different linear combinations of eigenfunctions, and still found the realignment happens mostly in the direction of the learned task, i.e., in all components. Interestingly, however, we have observed that the alignment tends to increase more for the first eigenfunctions in the mixture, and that the generalization tends to be worse when the mixture is composed of higher eigenfunctions.
>
> **Motivation of using alignment for convergence:** Indeed, in the kernel regime, hardness of generalization and optimization are tightly linked. As the main focus of our work was generalization, we decided to motivate this metric only from this point of view in the preliminaries with the aim to not clutter the main message of the paper. However, if the reviewer deemed it necessary, we could also motivate the connections to optimization (e.g. using [3]) in the appendix.
>
> **Effect of binarization:** Binarization has little effect on the direction of the eigenvectors in such high-dimensional setting, i.e., binarized and normal eigenvectors have an inner product of $0.78$ approximately (very large compared to $\sim10^{-4}$ for two random vectors). Moreover, when we tested all our observations using an $\ell_2$ loss and the standard eigenvectors we also observed the exact same phenomena.
>
> **Changes to Fig. 7:** We will make sure to better mention in the text that in the left plot alignment is equal to $\lambda^2$, and that in the right plot we use the original eigenvectors to compute the alignment.
>
> **Choice of K:** The energy concentration at the end of training grows very fast with increasing $K$ and saturates around $K=50$, that is why we chose such a number. It is true that we could have shown the evolution of the alignment instead of this metric. Note, however, that the absolute value of the alignment is heavily modulated by the magnitude of the eigenvalues of the network, and it is very hard to interpret without a proper reference. That is why we decided to show Fig. 6 in terms of energy concentration, as this gives a more clear picture of the evolution of the eigenspace, which complements the evolution of the spectrum in Fig. 7. In any case, we will include the plot for different values of $K$ in the appendix.

---

### Decision · Program_Chairs · 2021-09-27

**Decision:**

Accept (Poster)

**Comment:**

This paper explores what we can learn about neural network generalization through linearization, such as the empirical NTK. There were a number of supportive sentiments, but also concerns, discussed at length in the rebuttal period. One of the key concerns was novelty with respect to Baratin et. al. After discussion, it was understood that there are still novel contributions, such as the relative complexity of tasks being predicted with the empirical NTK, some experiments demonstrating the superiority of linearity model, and how alignment can hurt generalization. The final version of this paper should _carefully_ discuss Baratin et. al, and other related work, and address all of the reviewer questions, including experiments that were part of the rebuttal in the paper.